# Enhanced Optical Response of Zinc-Doped Tin Disulfide Layered Crystals Grown with the Chemical Vapor Transport Method

**DOI:** 10.3390/nano12091442

**Published:** 2022-04-23

**Authors:** Yu-Tai Shih, Der-Yuh Lin, Yu-Cheng Li, Bo-Chang Tseng, Sheng-Beng Hwang

**Affiliations:** 1Department of Physics, National Changhua University of Education, Changhua 500207, Taiwan; ytshih@cc.ncue.edu.tw; 2Department of Electronic Engineering, National Changhua University of Education, Changhua 500208, Taiwan; domo2048@gmail.com; 3Graduate Institute of Photonics, National Changhua University of Education, Changhua 500207, Taiwan; m1026007@gm.ncue.edu.tw; 4Department of Electronic Engineering, Chienkuo Technology University, Changhua 500020, Taiwan; sbhwa@ctu.edu.tw

**Keywords:** tin disulfide, zinc doping, photoelectric characteristics, chemical vapor transport method, layered material, photocurrent, photoresponsivity, optical response

## Abstract

Tin disulfide (SnS_2_) is a promising semiconductor for use in nanoelectronics and optoelectronics. Doping plays an essential role in SnS_2_ applications, because it can increase the functionality of SnS_2_ by tuning its original properties. In this study, the effect of zinc (Zn) doping on the photoelectric characteristics of SnS_2_ crystals was explored. The chemical vapor transport method was adopted to grow pristine and Zn-doped SnS_2_ crystals. Scanning electron microscopy images indicated that the grown SnS_2_ crystals were layered materials. The ratio of the normalized photocurrent of the Zn-doped specimen to that of the pristine specimen increased with an increasing illumination frequency, reaching approximately five at 10^4^ Hz. Time-resolved photocurrent measurements revealed that the Zn-doped specimen had shorter rise and fall times and a higher current amplitude than the pristine specimen. The photoresponsivity of the specimens increased with an increasing bias voltage or decreasing laser power. The Zn-doped SnS_2_ crystals had 7.18 and 3.44 times higher photoresponsivity, respectively, than the pristine crystals at a bias voltage of 20 V and a laser power of 4 × 10^−8^ W. The experimental results of this study indicate that Zn doping markedly enhances the optical response of SnS_2_ layered crystals.

## 1. Introduction

For more than a decade, research has focused on transition metal dichalcogenides (TMDCs) [1,2,3,4,5,6,7,8,9,10]. TMDCs have the general chemical formula MX_2_, where M represents a transition metal atom (group IV, V, VI, or VII) and X represents a chalcogen atom (S, Se, or Te). Many TMDCs are layered materials [11,12], in which each metal cation plane is between two chalcogen anion planes, which results in a sandwich-like X–M–X monolayer. These X–M–X monolayers are internally bonded through covalent bonding and held together through van der Waals (vdW) interactions. Because of the weak vdW forces, layered TMDCs can be readily cleaved into structures with few monolayers [13,14] or freestanding monolayers [15]. Because of their intriguing physical, chemical, and electronic properties, atomically thin two-dimensional (2D) TMDCs have attracted particular research attention and are regarded as promising candidates for use in catalysts, energy storage devices, electronic devices, biosensors and gas sensors, photonics devices, optoelectronic devices, and piezoelectric devices [6,8,16,17,18,19].

Tin disulfide (SnS_2_), which is an emerging post-transition metal dichalcogenide, is a layered material with a similar structure to TMDCs. In a SnS_2_ crystal, each Sn atom is covalently bonded to six S atoms through octahedral coordination to form S–Sn–S monolayers, and the monolayers are stacked through vdW forces. SnS_2_ has several polytypes [20,21], and two naming systems have been adopted to label these polytypes. Ramsdell’s notation [22] specifies the number of monolayers in the unit cell, followed by a letter to indicate the lattice type (T for tetragonal, H for hexagonal, or R for rhombohedral). Mitchell’s notation [23] specifies the number of S planes within the unit cell, followed by a letter (H or R) to indicate a hexagonal or rhombohedral symmetry. For example, the simplest possible polytype of SnS_2_ is labeled 1T using Ramsdell’s notation and 2H using Mitchell’s notation. In the present study, we used Mitchell’s system to name the common SnS_2_ polytypes, namely 2H and 4H SnS_2_. These polytypes have identical S–Sn–S monolayers but differ in terms of the stacking arrangement of the monolayers. The stacking arrangements of 2H and 4H SnS_2_ are [(AαB)]*_n_* and [(AαB)(CβB)]*_n_*, respectively, where the Roman letters represent S ions, and the Greek letters represent Sn ions [21].

SnS_2_ contains Earth-abundant elements and has environmentally benign characteristics. As a semiconducting material, SnS_2_ exhibits n-type characteristics [24,25,26,27] and has a sizable indirect bandgap in the range of 2.12–2.35 eV [28,29,30,31,32,33,34,35], which is wider than that of most TMDCs. The large bandgap benefits electronic applications, because it facilitates the restraint of source-to-drain tunneling in short-channel field-effect transistors (FETs) in integrated circuits [36,37,38]. Studies have been conducted on bulk and thin-film SnS_2_ to identify their optical absorption [29,39], reflectivity [28], energy band structure [40], photoemission [41], electronic charge density [42], Raman scattering [43], and dye sensitization [44] characteristics.

Research has been conducted on 2D SnS_2_ nanosheets and monolayers [26,32,33,38,45,46]. Sun et al. [45] were the first to synthesize SnS_2_ freestanding monolayers through a liquid exfoliation method. Their SnS_2_ monolayers were able to undergo visible-light water splitting with a high conversion efficiency of 38.7%. Song et al. [38] fabricated high-performance top-gated SnS_2_ monolayer FETs with a carrier mobility of 50 cm^2^/Vs and an on–off current ratio exceeding 10^7^. Huang et al. [32] characterized the properties of bulk, few-monolayer, and single-monolayer SnS_2_. They revealed that SnS_2_ has an indirect bandgap over its entire thickness range from the bulk material to a single monolayer. Ultrathin SnS_2_ transistors exhibit an on–off current ratio exceeding 10^6^ and a carrier mobility of up to 230 cm^2^/Vs. Gonzalez and Oleynik [46] used the first-principles density functional theory to investigate the layer-dependent structural, electronic, and vibrational properties of SnS_2_. They predicted a strong-layer dependence for the exciton binding energy and Raman intensity of 2D SnS_2_. Zhou et al. [33] synthesized large, ultrathin, single-crystalline SnS_2_ nanosheets with an improved chemical vapor deposition method. Their SnS_2_ nanosheet-based phototransistors exhibited high responsivities of 261 and 722 A/W in air and high vacuum, respectively. The aforementioned authors also fabricated a flexible photodetector based on SnS_2_ nanosheets, which demonstrated a high responsivity of 34.6 A/W. Thus, numerous studies have verified the potential of 2D SnS_2_ for use in nanoelectronics, optoelectronics, and energy conversion.

Doping plays a key role in research on SnS_2_, because it can increase the functionality of SnS_2_ by providing routes to tune its native properties [47,48,49,50,51,52,53,54,55,56]. For example, Zhou et al. [47] adopted a solvothermal method to synthesize 2D molybdenum (Mo)-doped SnS_2_ nanosheets for sensing nitrogen dioxide (NO_2_). They revealed that, for SnS_2_ nanosheets with a 3 at.% Mo doping concentration, the NO_2_ sensing response at 150 °C was enhanced by approximately 23 times relative to a pristine SnS_2_ specimen. Bouzid et al. [48] reported that a SnS_2_ single crystal with a 2 at.% cobalt doping concentration revealed a relatively high Curie temperature of approximately 131 K and a large saturation magnetization of approximately 0.65 emu g^−^^1^. Fan et al. [51] fabricated photodetectors based on indium (In)-doped few-monolayer SnS_2_. Compared with photodetectors based on pristine SnS_2_, the responsivity, external quantum efficiency, and normalized detectivity increased by up to two orders of magnitude after SnS_2_ was doped with 1.9 at.% of In. Meng et al. [52] synthesized aluminum (Al)-doped SnS_2_ nanosheets with a hydrothermal method. The response time and responsivity of a sample with a 6 at.% Al doping concentration were 20.4 and 19.2 times shorter and higher, respectively, than those of pristine SnS_2_. Lin et al. [53] used the first-principles calculations of the generalized gradient approximation method to study the magnetic and optical properties of 6.25 at.% chromium (Cr)-doped SnS_2_. Their calculation results revealed that, at approximately 1.17 eV, the absorption coefficient of 6.25 at.% Cr-doped SnS_2_ is 167,400 cm^−1^, which is considerably higher than that of gallium arsenide (40,000 cm^−1^), a commonly used absorption material in solar cells. Liu et al. [54] introduced sulfur vacancies into SnS_2_ nanostructures through copper (Cu) doping to improve the photocatalytic efficiency. They reported that the hydrogen generation rate of SnS_2_ doped with 5 at.% Cu reached 1.37 mmol h^−1^ g^−1^ under visible light, more than six times higher than that of pristine SnS_2_ nanoplates. Setayeshmehr et al. [55] synthesized alkali-metal-doped SnS_2_ nanostructures with a solvothermal method. Their sodium-doped SnS_2_ exhibited a high supercapacitor performance with a high capacitance of 269 Fg^−1^ at a current density of 1 Ag^−1^, approximately four times the specific capacitance of a pristine SnS_2_ nanostructure. Kumar et al. [56] studied hydrothermal synthesis zinc (Zn)-doped SnS_2_ nanoflakes at a low temperature (160 °C). Their experimental results revealed that Zn doping significantly improved the sensitivity of SnS_2_ to illumination. In summary, metal-doped SnS_2_ has excellent potential for use in sensing, hydrogen energy, energy storage, spintronic, and optoelectronic applications.

On the basis of the aforementioned studies, exploring the properties of metal-doped SnS_2_ crystals is warranted. Thus, in this study, pristine and Zn-doped SnS_2_ crystals were grown using the chemical vapor transport (CVT) method, and their morphological, structural, optical, and photoelectric properties were investigated. Our experimental results reveal that the grown SnS_2_ crystals formed layered materials, and their optical response was notably enhanced through Zn doping.

## 2. Materials and Methods

We adopted the CVT method to grow pristine and Zn-doped SnS_2_ crystals. An electronic balance was used to weigh high-purity Sn and S to generate a Sn:S molar ratio of 1:2. In addition, 0.3 g of iodine (I_2_) was adopted as a transport agent. Sn, S, and I_2_ were placed in a quartz ampoule along with the high-purity Zn doping element. The designed doping concentration was 2%. The quartz ampoule was evacuated to 1–2 × 10^−5^ Torr before being sealed and then placed in a three-zone furnace for 300 h. The temperature at one end of the quartz ampoule was set to 780 °C, and the temperature at the other end was set to 650 °C. The temperature gradient was approximately 4.3 °C/cm. The raw materials were initially located at the high-temperature end of the quartz ampoule. To obtain the optimal crystal quality, the temperatures of the two ends of the quartz ampoule were reversed once per day.

After the growth of the pristine and Zn-doped SnS_2_ crystals, a field-emission scanning electron microscope (S-4800, Hitachi, Tokyo, Japan) was used to characterize the morphology of the crystals. The chemical compositions of the specimens were identified using an energy dispersion X-ray spectroscope attached to the scanning electron microscope and a field-emission electron probe microanalyzer (JXA-8530F, JEOL, Tokyo, Japan). We employed a three-dimensional laser Raman microspectroscopy system (Nanofinder 30, Tokyo Instruments, Tokyo, Japan) equipped with a semiconductor laser with a wavelength of 488 nm to measure the Raman spectra of the crystals. The crystal images of the specimens were obtained using a transmission electron microscope (JEM-3010, JEOL, Tokyo, Japan). A high-resolution X-ray diffractometer (D8 DISCOVER SSS, Bruker, Billerica, MA, USA) that uses Cu Kα radiation (λ = 1.5418 Å) was adopted to examine the crystal structures of the specimens.

A 0.25 m monochromator (MKS, Irvine, CA, USA) equipped with a 130 W halogen lamp was used to produce monochromatic light with a wide photon energy range for the absorption, piezoreflectance (PzR), and photoconductivity (PC) measurements. We employed a mechanical chopper to modulate the continuous light from the monochromator into alternating incident light with a frequency of 200 Hz. For the PzR measurements, the measured specimen was attached to a piezoelectric ceramic holder, which was driven by a high-alternating-current (AC) voltage signal with a frequency of 200 Hz and a peak amplitude of 800 V to apply alternating stresses to the specimen. A silicon photodetector (Thorlabs, Newton, NJ, USA) with an amplifier was adopted to detect the intensity of the reflected light from the specimen’s surface. A dual-phase lock-in amplifier (Ametek, Berwyn, PA, USA) with the ability to suppress noise signals was used to record the output signals of the photodetector. For the absorption measurements, the measured specimen was attached to another holder. The silicon photodetector was placed on the back of the sample to detect the intensity of the transmitted light. For the PC measurements, a Keithley 2410 sourcemeter (Keithley, Solon, OH, USA) supplied a stable bias voltage of 20 V to the measured specimen. The photocurrent was recorded using a dual-phase lock-in amplifier and then divided by the power of the incident light at each wavelength to determine the photoresponsivity of the measured specimen.

To measure the photocurrent of a specimen as a function of the time or illumination frequency, a 520 nm wavelength laser was used as the excitation source. This laser was controlled by a function generator (3320A, Keysight, Singapore) to apply on–off light modulation to the measured specimen. In addition, a Keithley 2410 sourcemeter applied a stable bias voltage of 100 V to the measured specimen. For frequency-dependent photocurrent measurements, the photocurrent of the measured specimen under alternating illumination (*I*_ac_) was recorded using a dual-phase lock-in amplifier and then divided by the photocurrent under steady illumination (*I*_dc_) to obtain the normalized photocurrent (*I*_ac_/*I*_dc_) as a function of the alternating frequency of illumination. For time-dependent photocurrent measurements, a data acquisition device with a time resolution of 1 ns was used to collect photocurrent signals.

To measure the photoresponsivity of the measured specimen as a function of the laser power or bias voltage, we used a laser with a wavelength of 520 nm as the excitation source. A Keysight 3320 A function generator was used to modulate the laser light into alternating light with a frequency of 1 Hz. For laser-power-dependent photoresponsivity measurements, a rotary-vane-type variable attenuator, a neutral density (ND) 1.0 filter, and a ND 2.0 filter were used to adjust the laser power. A Keithley 2410 sourcemeter applied a stable voltage of 100 V to the measured specimen. The photocurrent was recorded using a dual-phase lock-in amplifier and divided by the laser power to determine the photoresponsivity of the measured specimen. For bias-voltage-dependent photoresponsivity measurements, we set the laser power to 1.29 mW, used a Keithley 2410 sourcemeter to apply a bias voltage to the measured specimen, and then recorded the induced current using a dual-phase lock-in amplifier.

## 3. Results and Discussion

Pristine and Zn-doped SnS_2_ crystals were grown using the CVT method. The thicknesses of the pristine and Zn-doped specimens were approximately 73 μm and 106 μm, respectively. The chemical compositions of the grown specimens were determined using an energy dispersive X-ray spectroscope (EDX) and a field-emission electron probe microanalyzer (EPMA). The atomic percentages of Sn and S in the pristine SnS_2_ crystals were 34.08% and 65.92%, respectively, when determined by the EDX, and 33.46% and 66.54%, respectively, when determined by the EPMA. The atomic percentages of Sn, S, and Zn in the Zn-doped SnS_2_ crystals were 34.31%, 65.32%, and 0.36%, respectively, when determined by the EDX, and 34.55%, 65.13%, and 0.31%, respectively, when determined by the EPMA. Each value is an average value calculated after multiple measurements; therefore, the sum of the atomic percentages of Sn, S, and Zn for the Zn-doped specimen is not exactly equal to 100%. The atomic ratio of Sn to S was approximately the ideal value of 1:2 for both specimens; however, the percentage of Zn was less than the expected value of 2% for the Zn-doped specimen.

Scanning electron microscopy (SEM) was used to observe the surface morphologies of the pristine and Zn-doped SnS_2_ crystals [Figure 1a,b]. The SEM images revealed that the grown SnS_2_ crystals were composed of multiple layers, and an angle of 120° characterized the edges of the layers. Figure 1c,d display the transmission electron microscopy (TEM) images of the pristine and Zn-doped SnS_2_ specimens, respectively. The insets are the selected area electron diffraction patterns of the corresponding SnS_2_ crystals. The images in Figure 1c,d depict a high-quality, single-crystalline hexagonal structure. The lattice plane spacing *d*_100_ of each SnS_2_ specimen was determined from its TEM image and is listed in Table 1. The lattice constant *a* of each SnS_2_ specimen was then calculated using the following formula [57]:(1)$$\frac{1}{d_{hkl}}=\sqrt{\frac{4}{3}\left(\frac{h^{2}+hk+k^{2}}{a^{2}}\right)+\frac{l^{2}}{c^{2}}},$$
and is also listed in Table 1. The calculated lattice constant *a* of the pristine SnS_2_ was 3.6812 Å, slightly larger than that reported by Palose et al. (Table 1) [58,59]. Marginal reductions in *d*_100_ and *a* were observed as Zn atoms were doped into the SnS_2_ crystals, possibly because the Zn ions replaced some of the Sn ions. The smaller radius of the Zn ions compared with that of the Sn ions resulted in smaller *d*_100_ and *a* values for the Zn-doped SnS_2_ crystals.

Raman spectroscopy was used to identify the polytype of the grown SnS_2_ crystals. The frequencies of the vibration modes of 2H and 4H SnS_2_ were reported by Smith et al. [43]. The room-temperature Raman spectra of the pristine and Zn-doped SnS_2_ layered crystals are presented in Figure 2a. These spectra indicate that the effect of Zn doping on the positions of the SnS_2_ Raman peaks was negligible. The frequency of the intense peaks (312.2 cm^−1^) was similar to the frequency of the *A*_1*g*_ optic mode of 2H SnS_2_ (315 cm^−1^) and the mixed *A*_1_ and *E* optic mode of 4H SnS_2_ (313.5 cm^−1^). Therefore, the polytype of the grown SnS_2_ crystals could not be identified by only using these intense peaks. However, Figure 2a also shows very weak peaks with frequencies of 200 and 214.4 cm^−1^. Smith et al. demonstrated that the *E* optic mode of 4H SnS_2_ is a doublet with frequencies of 200 and 214 cm^−1^, whereas the *E_g_* optic mode of 2H SnS_2_ is a singlet with a frequency of 205 cm^−1^. A doublet can be observed in Figure 2a; therefore, the grown crystals were 4H SnS_2_.

Figure 2b presents the X-ray diffraction patterns of the pristine and Zn-doped SnS_2_ layered crystals. Only the (00*l*) diffraction peaks of the SnS_2_ crystals can be observed in Figure 2b. The intense peak for the pristine specimen at 2*θ* = 15.00° corresponds to the (002) plane of the 4H SnS_2_ crystals, and the other weak peaks at 2*θ* = 30.28°, 46.20°, and 63.00° correspond to the (004), (006), and (008) planes, respectively. These peaks indicate that the [001] orientation was strongly preferred by the grown crystals. The grown SnS_2_ crystals had a CdI_2_-like layered structure belonging to the P6_3_*mc* space group. Their diffraction patterns matched well with those of the Joint Committee on Powder Diffraction Standards card No. 89-3198. Bragg’s diffraction formula is expressed as follows:(2)$$2d_{hkl}\sin\theta_{hkl}=n\lambda .$$

In this study, *λ* = 1.5418 Å (for the Cu Kα radiation); thus, by using Equation (2), the lattice constant *c* (=2*d*_002_) of the pristine SnS_2_ crystals was calculated to be 11.812 Å, which is consistent with that reported by Palose et al. (Table 1) [58,59]. For the Zn-doped specimen, Figure 2b shows peaks at 2*θ* = 15.00°, 30.37°, 46.13°, and 63.12°, corresponding to the (002), (004), (006), and (008) planes of the 4H SnS_2_ crystals, respectively. Therefore, Figure 2b reveals that the positions of the (00*l*) peaks of the Zn-doped SnS_2_ crystals are nearly the same as those of the pristine SnS_2_ crystals. Because the interactions between the S–Sn–S monolayers were weak vdW forces, the influence of Zn doping on the lattice constant *c* of the SnS_2_ layered crystals was minimal.

The absorption spectra of the pristine and Zn-doped SnS_2_ crystals were measured at room temperature to determine the optical bandgap. The optical absorption behavior of an indirect-bandgap semiconductor near the band edge can be expressed as follows [60,61,62]:(3)$$\alpha (E_{ph})\propto {\left[E_{ph}-(E_{g}\pm \hbar \Omega )\right]}^{2},$$
where *α* is the absorption coefficient, *E_ph_* is the energy of the incident photon, *E_g_* is the bandgap energy, and ℏΩ is the energy of a phonon being emitted (+ℏΩ) or absorbed (−ℏΩ). The absorbance *A* of a specimen is proportional to the absorption coefficient *α*, and in most situations, the energy of the phonon (ℏΩ) can be disregarded. Therefore, by using the Tauc plot (Figure 3a) [63], we obtained the bandgap by extrapolating the linear part of the *A*^1/2^ versus *E_ph_* curve at *A*^1/2^ = 0. The bandgap of the pristine SnS_2_ was 2.22 eV, which is consistent with that reported by Huang et al. for 4H SnS_2_ [32]. The bandgap of the Zn-doped SnS_2_ crystals was 2.30 eV. The uncertainty of these values was approximately 0.01 eV. As SnS_2_ was doped with Zn atoms, the bandgap of the SnS_2_ crystals increased. This increase might have resulted from the reduction in the lattice parameters *d*_100_ and *a*.

PC and PzR spectra can also be used to determine the bandgap. Figure 3b presents the PC spectra of the SnS_2_ crystals. The bandgaps of the pristine and Zn-doped SnS_2_ crystals were determined to be 2.22 and 2.30 eV, respectively, with an uncertainty of 0.01 eV. These values are the same as those indicated by the absorption spectra. Figure 3c depicts the PzR spectra of the SnS_2_ crystals. The bandgaps of the pristine and Zn-doped SnS_2_ crystals were determined to be 2.29 and 2.39 eV, respectively, with an uncertainty of 0.01 eV. These values are marginally higher than those indicated by the absorption and PC spectra.

Understanding the optical responsive properties of SnS_2_ layered crystals is essential when using them in optoelectronic devices. To investigate the dependency of photocurrents on the alternating frequency *f* of illumination, let *t*_i_ and *t*_d_ be the durations of the light and dark periods, respectively. For symmetric square light waves, *t*_i_ = *t*_d_ = *t*_0_ = 1/(2*f*). Let *τ* be the lifetime of carriers; if *t*_0_ ≫ *τ*, during a light interval, the photocurrent *I* increases with time as a function of *I*(*t*) = *I*_dc_(1 − *e^−t/τ^*) and finally reaches the steady-state value *I*_dc_. During a dark interval, in contrast, the photocurrent *I* decreases with time as a function of *I*(*t*) = *I*_dc_*e*^−*t*/*τ*^ and eventually vanishes. However, if *t*_0_ < τ, the photocurrent cannot reach the steady-state value during a light interval, nor can it reach 0 during a dark interval. After many light–dark cycles, the time average of the photocurrent becomes *I*_dc_/2. Let *I*_ac_ be the AC component of the photocurrent. The following equation is obtained:(4)$$\frac{I_{\mathrm{dc}}}{2}-\frac{I_{\mathrm{ac}}}{2}=\left(\frac{I_{\mathrm{dc}}}{2}+\frac{I_{\mathrm{ac}}}{2}\right)e^{-t_{0}/\tau }.$$

Therefore, by rearranging, the following equation is obtained [64]:(5)$$\frac{I_{\mathrm{ac}}}{I_{\mathrm{dc}}}=\frac{1-e^{-t_{0}/\tau }}{1+e^{-t_{0}/\tau }}=\tanh\left(\frac{t_{0}}{2\tau }\right)=\tanh\left(\frac{1}{4f\tau }\right).$$

A material can have more than one carrier-depleting process. If two processes are dominant, Equation (5) can be modified as follows:(6)$$\frac{I_{\mathrm{ac}}}{I_{\mathrm{dc}}}=c_{1}\tanh\left(\frac{1}{4f\tau_{1}}\right)+c_{2}\tanh\left(\frac{1}{4f\tau_{2}}\right).$$

In Equation (6), *c*_1_ and *c*_2_ are the proportional coefficients, and *τ*_1_ and *τ*_2_ are the carrier lifetimes for long-lifetime and short-lifetime decay processes, respectively.

Figure 4 illustrates the normalized photocurrent (*I*_ac_/*I*_dc_) of the pristine and Zn-doped SnS_2_ layered crystals as a function of the alternating frequency of illumination. The normalized photocurrent of the Zn-doped SnS_2_ crystals decreased more slowly than that of the pristine SnS_2_ crystals as the frequency increased. Therefore, the ratio of the normalized photocurrent of the Zn-doped SnS_2_ crystals to that of the pristine SnS_2_ crystals increased with an increasing alternating frequency, reaching 4.93 at 10^4^ Hz. When operated at a high alternating frequency, the optical response of the Zn-doped SnS_2_ crystals was superior to that of the pristine SnS_2_ crystals.

The frequency-dependent behavior of the photocurrent shown in Figure 4 can be fitted by Equation (6). The obtained values of the fitting parameters are listed in Table 2. For the pristine SnS_2_ crystals, 70% and 30% of the photocurrent can be attributed to long- and short-lifetime carriers, respectively. The proportion of the photocurrent attributed to short-lifetime carriers was higher for the Zn-doped SnS_2_ crystals than for the pristine SnS_2_ crystals, possibly because of the additional trap states produced by the Zn atoms, which can induce short-lifetime decay processes.

Figure 5a,b illustrate the time-dependent photocurrents of the pristine and Zn-doped SnS_2_ specimens and indicate how the photocurrent of each specimen changed over time under an alternating illumination frequency of 1000 Hz. The photocurrents of the specimens exhibited similar behaviors under other illumination frequencies. Table 3 lists the rise time *t*_rise_ (from 10% to 90% of the maximum photocurrent) and fall time *t*_fall_ (from 90% to 10% of the maximum photocurrent) for each specimen under different illumination frequencies. The rise and fall times of the Zn-doped SnS_2_ crystals were shorter than those of the pristine SnS_2_ crystals under all illumination frequencies. The current amplitude, which was defined as the difference between the maximum and minimum photocurrents in a rising–falling cycle, of each specimen under different illumination frequencies is listed in Table 4. Under all illumination frequencies, the current amplitude of the Zn-doped SnS_2_ crystals was higher than that of the pristine SnS_2_ crystals. According to the data listed in Table 3 and Table 4, Zn doping enhanced the response of the grown SnS_2_ crystals to light.

Figure 6a illustrates the photoresponsivity of each specimen as a function of the bias voltage. As the applied bias voltage increased, the photoresponsivity of each specimen gradually increased. The photoresponsivity of the Zn-doped SnS_2_ crystals was higher than that of the pristine SnS_2_ crystals at all bias voltages. At a bias voltage of 20 V, the photoresponsivity of the Zn-doped SnS_2_ crystals reached a maximum value of 30 μA/W, which was 7.18 times higher than that of the pristine SnS_2_ crystals, namely 4.18 μA/W.

Figure 6b depicts how the photoresponsivity of each specimen varied with the incident laser power. As the laser power gradually decreased from an order of 10^−3^ W to an order of 10^−8^ W, the photoresponsivity of each specimen steadily increased. This increase reached three orders of magnitude. For a given incident laser power, the photoresponsivity of the Zn-doped SnS_2_ crystals was higher than that of the pristine SnS_2_ crystals. At a laser power of 4 × 10^−8^ W, the photoresponsivity of the Zn-doped SnS_2_ crystals reached a maximum value of 8.04 mA/W, which was 3.44 times higher than that of the pristine SnS_2_ crystals, namely 2.34 mA/W.

## 4. Conclusions

In conclusion, pristine and Zn-doped SnS_2_ crystals were grown in this study with the CVT method, and their morphological, structural, optical, and photoelectric properties were studied. The SEM images revealed that the SnS_2_ crystals were layered materials, with an angle of 120° characterizing the edge of each layer. The doublet *E* mode and mixture *A*_1_ + *E* mode signals in the Raman spectra verified that the grown layered crystals were the 4H polytype of SnS_2_. The TEM results revealed that the lattice constant *a* of the pristine SnS_2_ crystals was approximately 3.681 Å. The value of parameter *a* reduced slightly as Zn atoms were doped into the SnS_2_ crystals, possibly because the Zn ions replaced some of the Sn ions. The X-ray diffraction results indicated that the lattice constant *c* of the pristine and Zn-doped SnS_2_ layered crystals was 11.812 Å. Because the interactions between the S–Sn–S monolayers were weak vdW forces, the influence of Zn doping on the lattice constant *c* was minimal. Moreover, the bandgap of the pristine SnS_2_ crystals was determined to be 2.22 eV by using absorption and PC spectra. After doping with Zn atoms, the bandgap increased. Frequency-dependent photocurrent measurements revealed that the normalized photocurrent of the Zn-doped SnS_2_ crystals decreased more slowly than that of the pristine SnS_2_ crystals as the frequency increased. When operated at a high alternating illumination frequency, the optical response of the Zn-doped SnS_2_ crystals was superior to that of the pristine SnS_2_ crystals. The time-dependent photocurrent measurements for the SnS_2_ layered crystals indicated that, under all illumination frequencies, the rise and fall times of the Zn-doped SnS_2_ crystals were shorter than those of the pristine SnS_2_ crystals, whereas the current amplitude of the Zn-doped SnS_2_ crystals was higher than that of the pristine SnS_2_ crystals. Moreover, experiments on laser-power-dependent and bias-voltage-dependent photoresponsivity revealed that Zn doping increased the photoresponsivity of SnS_2_. All of the experimental results indicate that Zn doping markedly enhances the optical response of SnS_2_ layered crystals, which suggests that Zn-doped SnS_2_ has the potential for use in optoelectronic devices.

## Figures and Tables

**Figure 1 nanomaterials-12-01442-f001:**
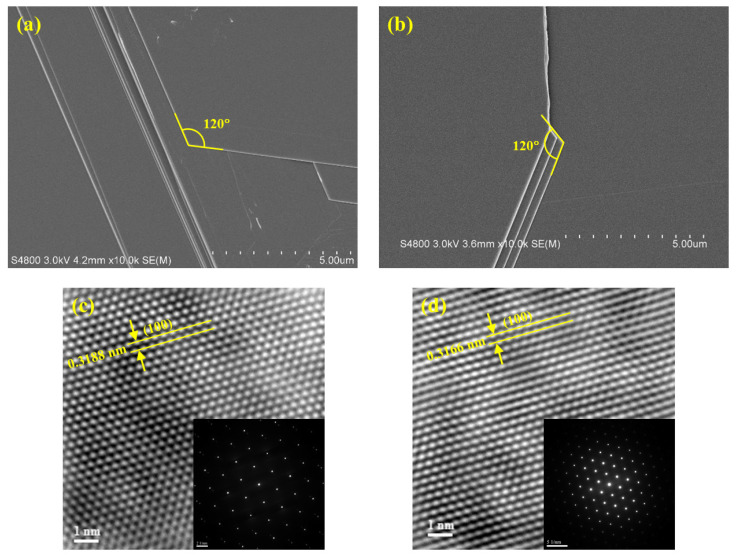
Scanning electron microscopy images of the (**a**) pristine and (**b**) Zn-doped SnS_2_ specimens. The measurement scale in each image represents a length of 5 μm (i.e., each division represents 0.5 μm). Transmission electron microscopy images of the (**c**) pristine and (**d**) Zn-doped SnS_2_ layered crystals are shown. The insets are the selected area electron diffraction patterns of the SnS_2_ layered crystals.

**Figure 2 nanomaterials-12-01442-f002:**
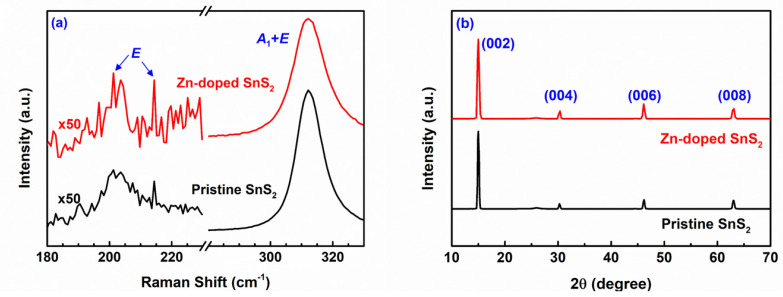
(**a**) Raman spectra and (**b**) X-ray diffraction patterns of the pristine and Zn-doped SnS_2_ layered crystals.

**Figure 3 nanomaterials-12-01442-f003:**
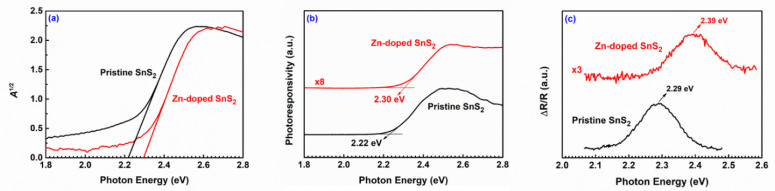
(**a**) Tauc plot of *A*^1/2^ (*A* represents the absorbance) versus the photon energy for the pristine and Zn-doped SnS_2_ layered crystals. (**b**) Photoconductivity and (**c**) piezoreflectance spectra of the pristine and Zn-doped SnS_2_ layered crystals.

**Figure 4 nanomaterials-12-01442-f004:**
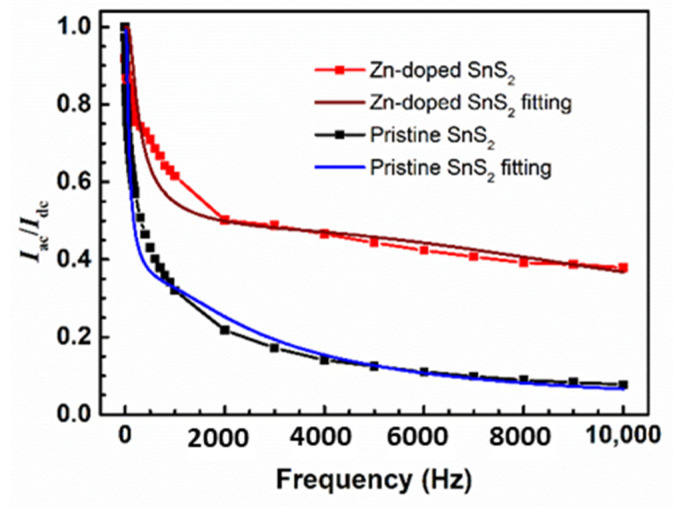
Normalized photocurrent of the pristine and Zn-doped SnS_2_ layered crystals as a function of the alternating frequency of illumination.

**Figure 5 nanomaterials-12-01442-f005:**
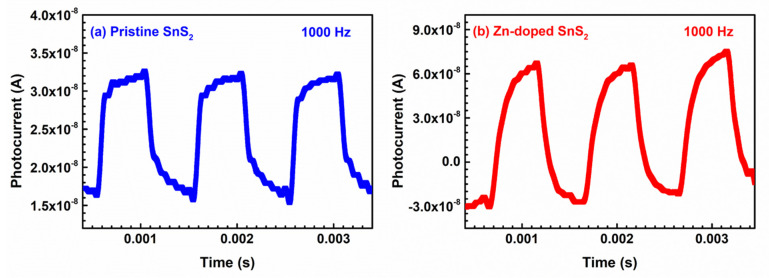
Photocurrents of the (**a**) pristine and (**b**) Zn-doped SnfS_2_ layered crystals as a function of time under an illumination frequency of 1000 Hz.

**Figure 6 nanomaterials-12-01442-f006:**
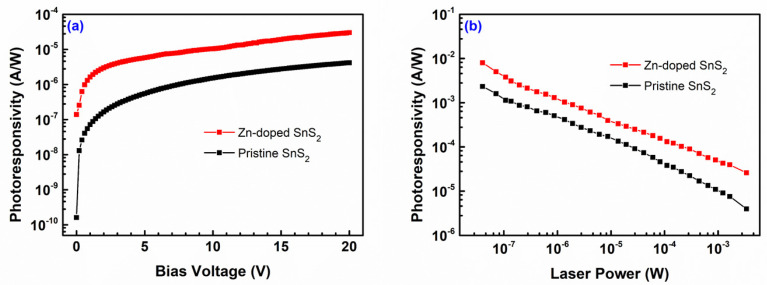
Photoresponsivity of the pristine and Zn-doped SnS_2_ layered crystals as a function of (**a**) the bias voltage and (**b**) the laser power.

**Table 1 nanomaterials-12-01442-t001:** Lattice parameters of the pristine and Zn-doped SnS_2_ layered crystals.

Specimen	*d*_100_ (Å)	*a* (Å)	*c* (Å)	References
2H SnS_2_		3.6470	5.8990	[58,59]
4H SnS_2_		3.6470	11.811	[58,59]
Pristine SnS_2_	3.1880	3.6812	11.812	This work
Zn-doped SnS_2_	3.1658	3.6556	11.812	This work

**Table 2 nanomaterials-12-01442-t002:** Obtained values for the fitting parameters used in Equation (6) for the pristine and Zn-doped SnS_2_ layered crystals.

Specimen	*c* _1_	*τ*_1_ (ms)	*c* _2_	*τ*_2_ (ms)
Pristine SnS_2_	0.70	4.96	0.30	0.119
Zn-doped SnS_2_	0.55	1.40	0.45	0.023

**Table 3 nanomaterials-12-01442-t003:** Rise time *t*_rise_ and fall time *t*_fall_ of the pristine and Zn-doped SnS_2_ layered crystals under different illumination frequencies.

	Frequency (Hz)							
	1		100		500		1000	
Specimen	*t*_rise_ (ms)	*t*_fall_ (ms)	*t*_rise_ (ms)	*t*_fall_ (ms)	*t*_rise_ (ms)	*t*_fall_ (ms)	*t*_rise_ (ms)	*t*_fall_ (ms)
Pristine SnS_2_	0.96	2.03	0.88	1.12	0.83	0.98	0.81	0.94
Zn-doped SnS_2_	0.31	0.25	0.45	0.23	0.22	0.21	0.21	0.19

**Table 4 nanomaterials-12-01442-t004:** Current amplitudes of the pristine and Zn-doped SnS2 layered crystals under different illumination frequencies.

	Frequency (Hz)			
	1	100	500	1000
Specimen	Current Amplitude (μA)			
Pristine SnS_2_	0.030	0.028	0.025	0.023
Zn-doped SnS_2_	0.110	0.100	0.090	0.080

## Data Availability

Data are contained within the article.

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
