# Peer review of "Enhanced Optical Response of Zinc-Doped Tin Disulfide Layered Crystals Grown with the Chemical Vapor Transport Method"

_nanomaterials, 2022, doi:10.3390/nano12091442_

Round 1

Reviewer 1 Report

The photoelectric properties of SnS2 and Zn-doped SnS2 have been explored. The authors have found enhanced optical response of SnS2 crystals.  In addition of growth properties, multiple characterizations have been evaluated. This work is interesting. It can be accepted after major revision.

Comments

  1. It has mentioned Zn-doped in SnS2. As I check XRD , there is shifting of peaks. It is incorporated in SnS2 crystal lattice or it just dopped. Can you provide EDX result to confirm its composition?
  2. Please include error bar in each bandgap estimation.
  3. Please include wider range of data in Fig.7.
  4. Please improve the fitting in Fig. 8.
  5. If you agree, please merge Fig. 1 and 3; Fig. 2 and 4; Fig. 5,6,7, and 8 etc. It has many single figures which could come under same category.

Author Response

Response to Reviewer 1

Comments and Suggestion

The photoelectric properties of SnS2 and Zn-doped SnS2 have been explored. The authors have found enhanced optical response of SnScrystals.  In addition of growth properties, multiple characterizations have been evaluated. This work is interesting. It can be accepted after major revision.

Comments

  1. It has mentioned Zn-doped in SnS2. As I check XRD , there is shifting of peaks. It is incorporated in SnS2 crystal lattice or it just dopped. Can you provide EDX result to confirm its composition?

Response:

Thank you very much for your extremely valuable comments and suggestions. Please see the new Fig. 2(b), Lines 267-269, and Lines 276-278 in our revised manuscript. The XRD peaks for the pristine SnS2 are at 2θ = 15.00°, 30.28°, 46.20°, and 63.00° corresponding to the (002), (004), (006), and (008) planes, respectively. The XRD peaks for the Zn-doped SnS2 are at 2θ = 15.00°, 30.37°, 46.13°, and 63.12°. The positions of the (00l) peaks of the Zn-doped SnS2 crystals are very close to those of the pristine SnS2 crystals.

We have added the EDX results in our revised manuscript. Please see Lines 201-205 of our revised manuscript. The atomic percentages of Sn and S in the pristine SnS2 crystals determined by the EDX are 34.08% and 65.92%, respectively. The atomic percentages of Sn, S, and Zn in the Zn-doped SnS2 crystals were 34.31%, 65.32%, and 0.36%, respectively. These values are very close to those values determined by the EPMA.

  1. Please include error bar in each bandgap estimation.

Response:

The uncertainty of each estimated bandgap has been stated in Lines 302-303, 308-309, and 311-312 of the revised manuscript.

  1. Please include wider range of data in Fig.7.

Response:

We have widened the range of data in Fig. 7. Please see the new Fig. 3(c) of our revised manuscript.

  1. Please improve the fitting in Fig. 8.

Response:

We have improved the fitting in Fig. 8. Please see the new Fig. 4 and Table 2 of our revised manuscript.

  1. If you agree, please merge Fig. 1 and 3; Fig. 2 and 4; Fig. 5,6,7, and 8 etc. It has many single figures which could come under same category.

Response:

The old Fig. 1 and 3 have been merged as the new Fig. 1. The old Fig. 2 and 4 have been merged as the new Fig. 2. The old Fig. 5, 6, and 7 have been merged as the new Fig. 3. The old Fig. 10 and 11 have been merged as the new Fig. 6.

Reviewer 2 Report

The authors of this manuscript have grown pristine and zinc doped SnS2 using chemical vapor transoport (CVT) method and investigated  their morphological, structural, optical, and photoelectric properties. The content of this manuscript is of scientific interest and within the scope of the journal. However, the authors may be advised to update their manuscript addressing the following comments.

  1. The title of the manuscript is quite broad since there are other published articles on Zoinc doped Tin Disulfide and their characterization including optical properties (https://doi.org/10.3390/nano9070924). The authors may highlight their novelty mentioning their method (CVT) of growing the crystals.
  2. In the introduction, the authors cited several articles on doped SnS2 but, they did not cite any articles on zinc doped SnS2. There are several such articles available online; one such article is cited in comment # 1.
  3. In the last paragraph of the introduction, the authors claimed, “To the best of our knowledge, the influence of zinc (Zn) doping 119 on the characteristics of SnS2 has not been comprehensively investigated.” The authors should avoid such claim to avoid any conflict with other published work.
  4. In the materials and method, the authors did not mention how long or how many days it took to grow crystals of SnS2 via CVT. Also, they did not mention the doping reagent or materials that is, the Zn doping element.
  5. What were the thickness of the samples?
  6. The authors should cite appropriate articles or book for equation 1.

Author Response

Response to Reviewer 2

Comments and Suggestion

The authors of this manuscript have grown pristine and zinc doped SnS2 using chemical vapor transoport (CVT) method and investigated their morphological, structural, optical, and photoelectric properties. The content of this manuscript is of scientific interest and within the scope of the journal. However, the authors may be advised to update their manuscript addressing the following comments.

The title of the manuscript is quite broad since there are other published articles on Zoinc doped Tin Disulfide and their characterization including optical properties (https://doi.org/10.3390/nano9070924). The authors may highlight their novelty mentioning their method (CVT) of growing the crystals.

Response:

Thank you very much for your extremely valuable comments and suggestions. We have changed the title of the manuscript. The new title is “Enhanced Optical Response of Zinc-Doped Tin Disulfide Layered Crystals Grown by Using Chemical Vapor Transport Method.”

In the introduction, the authors cited several articles on doped SnS2 but, they did not cite any articles on zinc doped SnS2. There are several such articles available online; one such article is cited in comment # 1.

Response:

The paper you mentioned (https://doi.org/10.3390/nano9070924) is of a great reference value. We have cited this paper as Ref. [56] in our manuscript. Please see Lines 116-119 of the revised manuscript.

In the last paragraph of the introduction, the authors claimed, “To the best of our knowledge, the influence of zinc (Zn) doping 119 on the characteristics of SnS2 has not been comprehensively investigated.” The authors should avoid such claim to avoid any conflict with other published work.

Response:

We have deleted the sentence “To the best of our knowledge, the influence of zinc (Zn) doping on the characteristics of SnS2 has not been comprehensively investigated.” Please see Lines 123-124 of the revised manuscript.

In the materials and method, the authors did not mention how long or how many days it took to grow crystals of SnS2 via CVT. Also, they did not mention the doping reagent or materials that is, the Zn doping element.

Response:

It took 300 hours to grow our SnS2 crystals via CVT method. Please see Lines 134-135 of the revised manuscript. We used the high-purity Zn element as the doping reagent. Please see Lines 132-133.

What were the thickness of the samples?

Response:

The thicknesses of the pristine and Zn-doped specimens were approximately 73 mm and 106 mm, respectively. Please see Lines 197-198.

The authors should cite appropriate articles or book for equation 1.

Response:

The book Elements of X-ray diffraction written by B. D. Cullity has been cited as Ref. [57] for Eq. 1. Please see Lines 220.

Round 2

Reviewer 1 Report

The revised manuscript has addressed all of comments made. I recommend it for publication.

Cheers